# Stereoselective Synthesis of 24-Fluoro-25-Hydroxyvitamin D_3_ Analogues and Their Stability to hCYP24A1-Dependent Catabolism

**DOI:** 10.3390/ijms222111863

**Published:** 2021-11-01

**Authors:** Fumihiro Kawagoe, Sayuri Mototani, Kaori Yasuda, Hiroki Mano, Toshiyuki Sakaki, Atsushi Kittaka

**Affiliations:** 1Faculty of Pharmaceutical Sciences, Teikyo University, 2-11-1 Kaga, Tokyo 173-8605, Japan; fkawagoe@pharm.teikyo-u.ac.jp (F.K.); 19dy10003vu@stu.teikyo-u.ac.jp (S.M.); 2Faculty of Engineering, Toyama Prefectural University, Imizu 939-0398, Japan; kyasuda@pu-toyama.ac.jp (K.Y.); z16003@st.pu-toyama.ac.jp (H.M.); tsakaki@pu-toyama.ac.jp (T.S.)

**Keywords:** human CYP24A1, synthesis, vitamin D_3_ metabolite, 24-fluoro-25-hydroxyvitamin D_3_ analogues, Sharpless dihydroxylation

## Abstract

Two 24-fluoro-25-hydroxyvitamin D_3_ analogues (**3**,**4**) were synthesized in a convergent manner. The introduction of a stereocenter to the vitamin D_3_ side-chain C24 position was achieved via Sharpless dihydroxylation, and a deoxyfluorination reaction was utilized for the fluorination step. Comparison between (24*R*)- and (24*S*)-24-fluoro-25-hydroxyvitamin D_3_ revealed that the C24-*R*-configuration isomer **4** was more resistant to CYP24A1-dependent metabolism than its 24*S*-isomer **3**. The new synthetic route of the CYP24A1 main metabolite (24*R*)-24,25-dihydroxyvitamin D_3_ (**6**) and its 24*S*-isomer (**5**) was also studied using synthetic intermediates (**30**,**31**) in parallel.

## 1. Introduction

Vitamin D_3_ is a lipophilic vitamin, and hydroxylation steps promoted by the cytochrome P450 family are essential for both activation and deactivation pathways. In the deactivation step, human cytochrome P450 24A1 (hCYP24A1) is one of the main enzymes catalyzing hydroxylation at the C23 or C24 positions of the 25-hydroxyvitamin D_3_ [25(OH)D_3_] side-chain, and several subsequent hydroxylation steps lead to vitamin D_3_-23,26-lactone or calcitroic acid (Figure 1) [1,2,3,4,5,6,7,8].

Recently, we developed a new methodology to synthesize 23-fluorinated vitamin D_3_ analogues (1,2), and identified their unique biological activities (Figure 1). The 23*S*-fluorinated isomer (1) showed higher metabolic resistance against hCYP24A1 than its 23*R*-isomer (2) [9,10]. On the other hand, the 23*R*-isomer (2) showed a greater binding affinity for human vitamin D receptor (*h*VDR) than its 23*S* isomer and natural 25(OH)D_3_ (unpublished data). Encouraged by these results, we have been interested in 24-substituted vitamin D_3_ analogues, 24-fluoro-25-hydroxyvitamin D_3_ (3,4), to study elongation of the half-life time of 25(OH)D_3_ against CYP24A1-dependent metabolism [11].

There have been several reports on the synthesis of 24-fluorinated vitamin D_3_ analogues. For example, in 1979, a 24-fluorinated vitamin D_3_ analogue was first reported by Ikekawa et al. [12,13]; they described 24-fluoro-25-hydroxyvitamin D_3_ (7) as a C24 diastereomeric mixture (Figure 2). Later, Uskoković et al. synthesized (24*R*)-24-fluoro-1α,25-dihydroxyvitamin D_3_ (**8**) from a steroid skeleton in 1985 and from a CD-ring fragment in 1988 [14,15]. However, selective synthesis of the 24*S*-fluorinated vitamin D_3_ analogue has not been reported, and the route to synthetic modification at C24 is still limited. Considering the importance of the C24 position of vitamin D_3_—including its stereochemistry—the practical synthetic methodology for 24-fluorinated vitamin D_3_ analogues is an essential topic.

To solve the problems above, we herein report a new stereoselective synthetic methodology for 24-fluoro-25-hydroxyvitamin D_3_ (**3**,**4**) through the chiral CD-ring part of 24,25-dihydroxyvitamin D_3_ (**5**,**6**), and reveal their preliminary biological activities. We considered that 24-substituted CD-ring fragments (**13**–**16**) may be useful units to synthesize numerous 24-substituted vitamin D_3_ analogues if coupled with various A-ring fragments [16] (Figure 2).

Synthesis of CD-ring fragments was achieved by side-chain elongation of Inhoffen–Lythgoe diol. Stereoselective introduction of the 24-hydroxy group was performed by Sharpless dihydroxylation reaction [17,18], and the fluorination step was achieved by deoxyfluorination reaction using *N*,*N*-diethylaminosulfur trifluoride (DAST).

## 2. Results and Discussion

For the synthesis of C24-substituted CD-ring fragments (**13**–**16**), commercially available Inhoffen–Lythgoe diol was chosen as a starting material (Figure 3). Iodination at C22-OH and hydroxy protection at the C8 position yielded iodide 18 [19]. After replacement of iodine with an allyl group utilizing allyl magnesium bromide, stereoselective dihydroxylation was achieved via Sharpless asymmetric dihydroxylation using AD-mix α and β to yield diols with 24*S*-OH (**20**) and 24*R*-OH (**21**), respectively. Protection of the C24 position with benzyl ether and two-step oxidation afforded carboxylic acids (**28**,**29**). These were treated with trimethylsilyl diazomethane in methanol to produce methyl esters (**30**,**31**), which were subsequently hydrogenated to afford 24-hydroxylated methyl esters (**9**,**10**). Next, introduction of a fluorine atom was achieved via deoxyfluorination reaction using DAST. The addition of an excess of methyl magnesium chloride to the resulting fluoro methyl esters (**11**,**12**) in THF, followed by desilylation at the C8 position in the presence of *p*-toluenesulfonic acid, yielded 24-fluorinated CD-ring fragments (**15**,**16**).

Oxidation of 24-fluorinated CD-ring fragments (**15**,**16**) with tetrapropylammonium perruthenate (TPAP) in the presence of 4-methylmorpholine *N*-oxide in methylene chloride, followed by protection of the C25-hydroxy group utilizing trimethylsilyl chloride (TMSCl), yielded 8-ketones (**32**,**33**) (Figure 4). The Wittig–Horner coupling reaction with the lithium salt of the A-ring phosphine oxide [16] produced the coupling products. The final deprotection with tetrabutylammonium fluoride (TBAF) afforded the desired 24-fluoro-25-hydroxyvitamin D_3_ (**3** and **4**) in 50 and 61% overall yields from **15** and **16**, respectively.

There are several methods to synthesize 24-hydroxyvitamin D_3_ analogues [20,21,22,23,24,25,26,27]. In this study, we also explored the possibility of using 24-*O*-benzyl methyl esters (30,31) to synthesize their important precursors (13,14). As shown in Figure 5, the 24-*O*-benzyl methyl esters were subsequently reacted with methyl magnesium chloride to produce 34 and 35. Deprotection of the benzyl group afforded 36 and 37, respectively, and desilylation at the C8-OH with *p*-toluenesulfonic acid yielded 24-hydroxy CD-rings (13,14).

To construct triene structures, we took advantage of a method that Sarandeses et al. developed in 2002 [25]. 24,25-Diol protection of the 24-hydroxylated CD-ring fragments (13,14) as a ketal was performed with 2,2-dimethoxypropane in the presence of pyridinium *p*-toluenesulfonate (PPTS) as an acid catalyst, and subsequent oxidation with TPAP and NMO of C8-hydroxy groups afforded the desired 8-ketones (40,41). The coupling reaction between the CD-rings (40,41) and A-ring phosphine oxide [16] was performed via the Wittig–Horner reaction to yield the protected vitamin D_3_. Deprotection with TBAF followed by cationic exchange resin (AG 50W-X4, H^+^ form) treatment afforded 24,25-dihydroxyvitamin D_3_ (**5**,**6**).

### Biological Evaluation

The binding affinities of the three 24-fluorinated vitamin D_3_ analogues—(24*S*)-24-F-25(OH)D_3_ (**3**), (24*R*)-24-F-25(OH)D_3_ (**4**), and 24,24-difluoro-25(OH)D_3_ [28]—for hVDR are summarized in Table 1. For hVDR, **3** and **4** showed similar binding affinities, but slightly lower than that of natural 25(OH)D_3_. These results demonstrate that a fluorine atom at the C24 position could mildly impair the binding with hVDR. However, unexpectedly, 24,24-difluoro-25(OH)D_3_ showed higher binding affinity for hVDR than those of the 24-fluorinated vitamin D_3_ analogues **3** and **4**.

We next analyzed the metabolism of three analogues and 25(OH)D_3_ by hCYP24A1. Hydroxylation activities of hCYP24A1 toward these analogues are shown in Table 2. The hCYP24A1 showed nearly the same activity toward (24*S*)-24-F-25(OH)D_3_ as that toward 25(OH)D_3_, whereas 24,24-F_2_-25(OH)D_3_ showed marked resistance to hCYP24A1-dependent metabolism. These results demonstrate that the 24*R* fluorine substitution allows 25(OH)D_3_ to achieve stronger catabolic resistance than its 24*S* counterpart. In contrast, we demonstrated that (23*S*)-23-F-25(OH)D_3_ (**1**) showed stronger resistance to CYP24A1 metabolism than (23*R*)-23-F-25(OH)D_3_ (**2**), as described in our previous study [9]. These results can be explained by the direction of hydroxylation at the C23 and C24 positions by CYP24A1 [1,2,3,4,5,6,7,8].

## 3. Experimental Section

^1^H and ^13^C NMR spectra were recorded on JEOL AL-400 NMR (400 MHz) and ECP-600 NMR (600 MHz) spectrometers (Tokyo, Japan). ^1^H NMR spectra were referenced with (CH_3_)_4_Si (δ 0.00 ppm) or CHCl_3_ (δ 7.26 ppm) as internal standards. ^13^C NMR spectra were referenced with deuterated solvent (δ 77.0 ppm for CDCl_3_). IR spectra were recorded on a JASCO FT-IR-800 Fourier-transform infrared spectrophotometer (Tokyo, Japan). High-resolution mass spectra were obtained on a SHIMADZU LCMS-IT-TOF mass spectrometer (Kyoto, Japan) with an electrospray ionization (ESI) method or atmospheric-pressure chemical ionization (APCI). Optical rotations were measured on a JASCO DIP-370 digital polarimeter (Tokyo, Japan). Column chromatography was performed on silica gel 60N (40–50 μm, Kanto Chemical Co., Inc., Tokyo, Japan) or silica gel 60 (0.040–0.063 mm, Merck, Tokyo Japan). All experiments were performed under anhydrous conditions in an atmosphere of argon, unless otherwise stated. The supporting information of ^1^H and ^13^C NMR spectra of all new compounds: **19**–**21**, **24**, **25**, **28**–**31**, **9**–**12**, **15**, **16**, **3**, **4**, **36**, and **37** is available at the link in Appendix A.

### 3.1. tert-Butyl({(1R,3aR,4S,7aR)-1-[(R)-hex-5-en-2-yl]-7a-methyloctahydro-1H-inden-4-yl}oxy) Dimethylsilane *(**19**)*

To a solution of compound **18 [19]** (180.0 mg, 0.412 mmol) in THF (4 mL), allyl magnesium bromide (3.3 mL, 1 M in Et_2_O, 3.3 mmol) was added at 0 °C, and it was stirred at room temperature for 23 h. After the reaction was quenched with water and aqueous saturated NH_4_Cl, the mixture was extracted with EtOAc three times, dried over Na_2_SO_4_, filtered, and concentrated. The residue was purified via flash column chromatography on silica gel (hexane only) to obtain **19** (105.1 mg, 73%) as a colorless oil.

**19**: [α]_D_^27^ +52.7 (c 1.82, CHCl_3_); IR (neat) 1471, 1371, 1252, 1162, 1085, 1027, 837, 771 cm^−1^; ^1^H NMR (600 MHz, CDCl_3_) δ −0.01 (s, 3H), 0.01 (s, 3H), 0.89 (s, 9H), 0.90 (d, *J* = 6.6 Hz, 3H), 0.91 (s, 3H), 0.99–1.13 (m, 3H), 1.21–1.28 (m, 2H), 1.30–1.43 (m, 4H), 1.46–1.58 (m, 2H), 1.63–1.70 (m, 1H), 1.74–1.84 (m, 2H), 1.90–1.97 (m, 2H), 2.08–2.14 (m, 1H), 3.99–4.00 (m, 1H), 4.90–4.92 (m, 1H), 4.97–5.04 (m, 1H), 5.80 (tdd, *J* = 6.0, 10.2, 16.2 Hz, 1H); ^13^C NMR (150 MHz, CDCl_3_) δ −5.2, −4.8, 13.7, 17.7, 18.0, 18.5, 23.1, 25.8, 27.3, 30.5, 34.5, 34.9, 35.1, 40.7, 42.2, 53.1, 56.8, 69.5, 113.8, 139.7; HRMS (ESI^+^) calcd for C_22_H_42_OSi [M]^+^ 350.2999, found 350.2992.

### 3.2. (2S,5R)-5-{(1R,3aR,4S,7aR)-4-[(tert-Butyldimethylsilyl)oxy]-7a-methyloctahydro-1H-inden-1-yl}hexane-1,2-diol *(**20**)*

A mixture of AD-mix α (4.01 g) in *t*BuOH (10 mL) and H_2_O (10 mL) was stirred at 0 °C for 25 min; **19** (303.5 mg, 0.255 mmol) was added to the mixture at 0 °C, and it was stirred at the same temperature for 5 h, and then at room temperature for 15 h under air. After the reaction was quenched with water, the mixture was extracted with EtOAc three times, dried over Na_2_SO_4_, filtered, and concentrated. The residue was purified via flash column chromatography on silica gel (hexane:EtOAc = 1:1) to obtain **20** (253.9 mg, 79%) as a colorless oil.

**20**: [α]_D_^27^ +44.4 (c 1.55, CHCl_3_); IR (neat) 3402, 1645, 1469, 1374, 1265, 1160, 1066, 1032, 840, 776, 743 cm^−1^; ^1^H NMR (600 MHz, CDCl_3_) δ −0.01 (s, 3H), 0.00 (s, 3H), 0.88–0.90 (m, 15H), 0.98–1.12 (m, 3H), 1.20–1.43 (m, 7H), 1.47–1.58 (m, 3H), 1.64–1.67 (m, 1H), 1.75–1.83 (m, 2H), 1.92–1.95 (m, 1H), 2.27 (s, 3H), 3.41–3.44 (m, 1H), 3.62–3.67 (m, 2H), 3.98–3.99 (m, 1H); ^13^C NMR (150 MHz, CDCl_3_) δ −5.2, −4.8, 13.7, 17.6, 18.0, 18.6, 23.0, 25.8, 27.3, 29.7, 31.5, 34.4, 35.3, 40.7, 42.1, 53.0, 56.5, 66.7, 69.4, 73.0; HRMS (APCI^−^) calcd for C_22_H_44_O_3_SiCl [M+Cl]^−^ 419.2754, found 419.2764.

### 3.3. (2R,5R)-5-{(1R,3aR,4S,7aR)-4-[(tert-Butyldimethylsilyl)oxy]-7a-methyloctahydro-1H-inden- 1-yl}hexane-1,2-diol *(**21**)*

A mixture of AD-mix β (4.62 g) in *t*BuOH (15 mL) and H_2_O (15 mL) was stirred at 0 °C for 25 min; **19** (418.4 mg, 0.255 mmol) was added to the mixture at 0 °C, and it was stirred at the same temperature for 1 h 35 min under air. After the reaction was quenched with water, the mixture was extracted with EtOAc three times, dried over Na_2_SO_4_, filtered, and concentrated. The residue was purified via flash column chromatography on silica gel (hexane:EtOAc = 1:1) to obtain **21** (433.1 mg, 94%) as a colorless oil.

**21**: [α]_D_^27^ +41.9 (c 2.05, CHCl_3_); IR (neat) 3294, 1223, 1076, 1026, 837, 764 cm^−1^; ^1^H NMR (600 MHz, CDCl_3_) δ −0.01 (s, 3H), 0.00 (s, 3H), 0.88–0.90 (m, 15H), 1.01–1.58 (m, 11H), 1.65–1.67 (m, 1H), 1.75–1.84 (m, 2H), 1.91–1.95 (m, 4H), 3.42–3.45 (m, 1H), 3.64–3.69 (m, 2H), 3.99 (dd, *J* = 2.4, 5.4 Hz, 1H); ^13^C NMR (150 MHz, CDCl_3_) δ −5.2, −4.8, 13.7, 17.7, 18.0, 18.5, 23.0, 25.8, 27.3, 29.6, 31.4, 34.4, 35.1, 40.7, 42.1, 53.0, 56.5, 67.0, 69.4, 72.7; HRMS (ESI^−^) calcd for C_22_H_44_O_3_SiCl [M+Cl]^−^ 419.2754, found 419.2773.

### 3.4. (2S,5R)-2-(Benzyloxy)-5-{(1R,3aR,4S,7aR)-4-[(tert-butyldimethylsilyl)oxy]-7a-methyloctahy dro-1H-inden-1-yl}hexan-1-ol *(**24**)*

Benzaldehyde dimethyl acetal (374.4 mg, 369 μL, 2.46 mmol) and pyridinium *p*-toluenesulfonate (PPTS) (158.6 mg, 0.63 mmol) were added to a solution of **20** (472.0 mg, 1.23 mmol) in toluene (15 mL) at room temperature, and the mixture was stirred at the same temperature for 2 h. After the reaction was quenched with water and saturated aqueous NaHCO_3_, the mixture was extracted with EtOAc three times, dried over Na_2_SO_4_, filtered, and concentrated. The residue was purified via flash column chromatography on silica gel (hexane:EtOAc = 4:1) to obtain the crude acetal **22**, which was used for the next reaction without further purification. To a solution of the above crude acetal **22** in CH_2_Cl_2_ (15 mL), we added DIBAL-H (4.8 mL, 1.03 M in hexane solution, 4.92 mmol) at 0 °C, and the mixture was stirred at the same temperature for 20 min. After the reaction was quenched with MeOH at 0 °C, H_2_O and saturated aqueous potassium sodium tartrate were added at room temperature. The mixture was extracted with CH_2_Cl_2_ four times, dried over Na_2_SO_4_, filtered, and concentrated. The residue was purified via flash column chromatography on silica gel (hexane:EtOAc = 4:1) to obtain **24** (501.1 mg, 86%) as a colorless oil.

**24**: [α]_D_^27^ +49.0 (c 3.62, CHCl_3_); IR (neat) 3420, 1465, 1453, 1374, 1254, 1085, 1028, 840, 776, 739 cm^−1^; ^1^H NMR (600 MHz, CDCl_3_) δ 0.00 (s, 3H), 0.01 (s, 3H), 0.90–0.91 (m, 15H), 0.99–1.13 (m, 3H), 1.21–1.60 (m, 10H), 1.66–1.68 (m,1H), 1.75–1.84 (m, 2H), 1.94–1.96 (m, 2H), 3.45–3.48 (m, 1H), 3.51–3.54 (m, 1H), 3.69 (dd, *J* = 3.0, 11.4 Hz, 1H), 4.00–4.00 (m, 1H), 4.53 (d, *J* = 11.7 Hz, 1H), 4.63 (d, *J* = 11.7 Hz, 1H), 7.28–7.32 (m, 1H), 7.35–7.36 (m, 4H); ^13^C NMR (150 MHz, CDCl_3_) δ −5.2, −4.8, 13.7, 17.6, 18.0, 18.5, 23.0, 25.8, 27.1, 27.3, 31.2, 34.4, 35.3, 40.7, 42.1, 53.0, 56.5, 64.2, 69.4, 71.4, 80.4, 127,7, 127.8, 128.4, 138.5; HRMS (ESI^+^) calcd for C_29_H_50_NaO_3_Si [M + Na]^+^ 497.3421, found 497.3433.

### 3.5. (2R,5R)-2-(Benzyloxy)-5-{(1R,3aR,4S,7aR)-4-[(tert-butyldimethylsilyl)oxy]-7a-methyloctahydro-1H-inden-1-yl}hexan-1-ol *(**25**)*

Benzaldehyde dimethyl acetal (39.3 mg, 39 μL, 0.258 mmol) and pyridinium *p*-toluenesulfonate (PPTS) (4.5 mg, 0.018 mmol) were added to a solution of **21** (49.6 mg, 0.129 mmol) in toluene (0.7 mL) at room temperature, and the mixture was stirred at the same temperature for 2 h. After the reaction was quenched with water and saturated aqueous NaHCO_3_, the mixture was extracted with EtOAc three times, dried over Na_2_SO_4_, filtered, and concentrated. The residue was purified via flash column chromatography on silica gel (hexane:EtOAc = 5:1) to obtain the crude acetal **23**, which was used for the next reaction without further purification. To a mixture of the above crude acetal **23** in CH_2_Cl_2_ (2 mL), we added DIBAL-H (313 μL, 1.03 M in hexane solution, 0.322 mmol) at −40 °C, and the mixture was stirred at the same temperature for 1 h, and then at room temperature for 1 h. After the reaction was quenched with MeOH, H_2_O and saturated aqueous potassium sodium tartrate were added at room temperature. The mixture was extracted with CH_2_Cl_2_ three times, dried over Na_2_SO_4_, filtered, and concentrated. The residue was purified via flash column chromatography on silica gel (hexane:EtOAc = 4:1) to obtain **25** (44.4 mg, 72%) as a colorless oil.

**25**: [α]_D_^27^ +31.1 (c 0.91, CHCl_3_); IR (neat) 3332, 1462, 1369, 1257, 1076, 1030, 837, 771 cm^−1^; ^1^H NMR (600 MHz, CDCl_3_) δ 0.00 (s, 3H), 0.01 (s, 3H), 0.89–0.90 (m, 15H), 1.00–1.14 (m, 3H), 1,20–1.26 (m, 2H), 1.30–1.45 (m, 6H), 1.51–1.58 (m, 2H), 1.65–1.71 (m, 2H), 1.75–1.84 (m, 2H), 1.93–1.95 (m, 1H), 3.46–3.50 (m, 1H), 3.52–3.55 (m, 1H), 3.68 (dd, *J* = 3.0, 12.0 Hz, 1H), 3.99–4.00 (m, 1H), 4.54 (d, *J* = 10.8 Hz, 1H), 4.62 (d, *J* = 10.8 Hz, 1H), 7.28–7.32 (m, 1H), 7.35–7.36 (m, 4H); ^13^C NMR (150 MHz, CDCl_3_) δ −5.2, −4.8, 13.7, 17.7, 18.0, 18.6, 23.0, 25.8, 27.2, 27.3, 31.1, 34.4, 35.3, 40.7, 42.1, 53.0, 56.4, 64.4, 69.4, 71.6, 80.3, 127,7, 127.8, 128.5, 138.5; HRMS (APCI^+^) calcd for C_29_H_50_NaO_3_Si [M + Na]^+^ 497.3421, found 497.3450.

### 3.6. (2S,5R)-2-(Benzyloxy)-5-({1R,3aR,4S,7aR)-4-[(tert-butyldimethylsilyl)oxy]-7a-methyloctahydro-1H-inden-1-yl}hexanoic acid *(**28**)*

Dess–Martin periodinane (1.42 g, 3.35 mmol) was added to a mixture of **24** (490.2 mg, 1.03 mmol) and 4Å molecular sieves (321.9 mg) in CH_2_Cl_2_ (10 mL) at 0 °C, and the mixture was stirred at the same temperature for 2 h. After the reaction was quenched with water and saturated aqueous NaHCO_3_, the mixture was extracted with CH_2_Cl_2_ three times, dried over Na_2_SO_4_, filtered, and concentrated. The residue was purified via flash column chromatography on silica gel (hexane:EtOAc = 10:1) to obtain the crude aldehyde **26**, which was used for the next reaction without further purification. To a mixture of the above crude aldehyde **26** and NaH_2_PO_4_ (1.216 g, 8.11 mmol) in H_2_O (9 mL) and *t*-BuOH (18 mL), NaClO_2_ (575.9 mg, 6.37 mmol) was added at 0 °C under air and stirred at the same temperature for 30 min. After the reaction was quenched with aqueous saturated NH_4_Cl and aqueous saturated sodium thiosulfate, the mixture was extracted with EtOAc three times, washed with brine, dried over Na_2_SO_4_, filtered, and concentrated. The residue was purified via flash column chromatography on silica gel (hexane:EtOAc = 1:1) to obtain **28** (960.6 mg, 99%) as a colorless oil.

**28**: [α]_D_^27^ +21.7 (c 1.32, CHCl_3_); IR (neat) 1720, 1469, 1254, 1089, 1032, 840, 780 cm^−1^; ^1^H NMR (400 MHz, CDCl_3_) δ −0.01 (s, 3H), 0.01 (s, 3H), 0.89–0.90 (m, 15H), 0.99–1.94 (m, 17H), 3,94–3.99 (m, 2H), 4.50 (d, *J* = 11.6 Hz, 1H), 4.70 (d, *J* = 11.6 Hz, 1H), 7.29–7.39 (m, 5H); ^13^C NMR (100 MHz, CDCl_3_) δ −5.2, −4.8, 13.7, 17.6, 18.0, 18.5, 23.0, 25.8, 27.2, 29.1, 31.1, 34.4, 35.0, 40.7, 42.1, 53.0, 56.4, 69.4, 72.5, 78.3, 128,1, 128.1, 128.5, 137.0, 176.7; HRMS (ESI^−^) calcd for C_29_H_47_O_4_Si [M-H]^−^ 487.3249, found 487.3278.

### 3.7. (2R,5R)-2-(Benzyloxy)-5-({1R,3aR,4S,7aR)-4-[(tert-butyldimethylsilyl)oxy]-7a-methyloctahydro-1H-inden-1-yl}hexanoic acid *(**29**)*

Dess–Martin periodinane (2.76 g, 6.51 mmol) was added to a mixture of **25** (1.03 g, 2.17 mmol) and 4Å molecular sieves (600.0 mg) in CH_2_Cl_2_ (10 mL) at 0 °C, and the mixture was stirred at the same temperature for 2 h. After the reaction was quenched with water and saturated aqueous NaHCO_3_, the mixture was extracted with CH_2_Cl_2_ three times, dried over Na_2_SO_4_, filtered, and concentrated. The crude residue **27** was used for the next reaction without further purification. To a mixture of the above crude aldehyde **27** in H_2_O (3 mL) and *t*-BuOH (6 mL), NaH_2_PO_4_ (134.8 mg, 0.898 mmol) and NaClO_2_ (24.6 mg, 0.272 mmol) were added at 0 °C under air and stirred at the same temperature for 30 min. After the reaction was quenched with aqueous saturated NH_4_Cl and aqueous saturated sodium thiosulfate, the mixture was extracted with EtOAc three times, washed with brine, dried over Na_2_SO_4_, filtered, and concentrated. The residue was purified via flash column chromatography on silica gel (hexane:EtOAc = 1:1) to obtain **29** (125.3 mg, quantitative yield) as a colorless oil.

**29**: [α]_D_^27^ +47.1 (c 1.88, CHCl_3_); IR (neat) 1720, 1469, 1250, 1085, 1028, 840, 776 cm^−1^; ^1^H NMR (400 MHz, CDCl_3_) δ 0.00 (s, 3H), 0.01 (s, 3H), 0.88–0.89 (m, 15H), 0.99–1.95 (m, 17H), 3,97–4.00 (m, 2H), 4.49 (d, *J* = 11.9 Hz, 1H), 4.71 (d, *J* = 11.9 Hz, 1H), 7.29–7.37 (m, 5H); ^13^C NMR (100 MHz, CDCl_3_) δ −5.2, −4.8, 13.7, 17.6, 18.0, 18.4, 23.0, 25.8, 27.2, 28.9, 30.6, 34.4, 34.7, 40.7, 42.1, 53.0, 56.3, 69.4, 72.6, 77.8, 128,1, 128.2, 128.5, 136.9, 176.2; HRMS (ESI^−^) calcd for C_29_H_47_O_4_Si [M-H]^−^ 487.3249, found 487.3269.

### 3.8. Methyl (2S,5R)-2-(Benzyloxy)-5-{(1R,3aR,4S,7aR)-4-[(tert-butyldimethylsilyl)oxy]-7a-methyloctahydro-1H-inden-1-yl}hexanoate *(**30**)*

Trimethylsilyl diazomethane (1.1 mL, 2.0 M in diethyl ether, 2.16 mmol) was added to a solution of **28** (490.2 mg, 1.03 mmol) in MeOH (2 mL) and CH_2_Cl_2_ (6 mL) at 0 °C, and the mixture was stirred at the same temperature for 17 min. After the reaction was quenched with acetic acid and saturated aqueous NaHCO_3_, the mixture was extracted with CH_2_Cl_2_ three times, dried over Na_2_SO_4_, filtered, and concentrated. The residue was purified via flash column chromatography on silica gel (hexane:EtOAc = 4:1) to obtain **30** (387 mg, 100%) as a colorless oil.

**30**: [α]_D_^27^ +17.0 (c 2.70, CHCl_3_); IR (neat) 1750, 1465, 1254, 1028, 840, 772 cm^−1^; ^1^H NMR (600 MHz, CDCl_3_) δ 0.01 (s, 3H), 0.01 (s, 3H), 0.88–0.89 (m, 15H), 1.01–1.10 (m, 3H), 1.19–1.26 (m, 2H), 1.30–1.42 (m, 2H), 1.50–1.67 (m, 4H), 1.73–1.85 (m, 3H), 1.91–1.94 (m, 1H), 3.75 (s, 3H), 3.89 (dd, *J* = 5.4, 7.8 Hz, 1H), 3.99–3.99 (m, 1H), 4.41 (d, *J* = 11.4 Hz, 1H), 4.68 (d, *J* = 11.4 Hz, 1H), 7.27–7.31 (m, 1H), 7.33–7.36 (m, 4H); ^13^C NMR (150 MHz, CDCl_3_) δ −5.2, −4.8, 13.7, 17.6, 18.0, 18.5, 23.0, 25.8, 27.1, 29.6, 31.2, 34.4, 35.0, 40.7, 42.1, 51.8, 53.0, 56.4, 69.4, 72.2, 78.9, 127.8, 127.9, 128.3, 137.6, 173.4; HRMS (ESI^+^) calcd for C_30_H_49_O_4_SiNa [M + Na]^+^ 525.3371, found 525.3389.

### 3.9. Methyl (2R,5R)-2-(Benzyloxy)-5-{(1R,3aR,4S,7aR)-4-[(tert-butyldimethylsilyl)oxy]-7a-methyloctahydro-1H-inden-1-yl}hexanoate *(**31**)*

Trimethylsilyl diazomethane (362 μL, 2.0 M in diethyl ether, 0.73 mmol) was added to a solution of **29** (125.3 mg, 1.03 mmol) in MeOH (1.5 mL) and CH_2_Cl_2_ (4.5 mL) at 0 °C, and the mixture was stirred at the same temperature for 20 min. After the reaction was quenched with acetic acid and saturated aqueous NaHCO_3_, the mixture was extracted with CH_2_Cl_2_ three times, dried over Na_2_SO_4_, filtered, and concentrated. The residue was purified via flash column chromatography on silica gel (hexane:EtOAc = 4:1) to obtain **31** (125.1 mg, 97%) as a colorless oil.

**31**: [α]_D_^27^ +57.5 (c 1.71, CHCl_3_); IR (neat) 1750, 1471, 1253, 1029, 838, 774 cm^−1^; ^1^H NMR (400 MHz, CDCl_3_) δ 0.00 (s, 3H), 0.01 (s, 3H), 0.86–0.89 (m, 15H), 1.90–1.95 (m, 1H), 3.75 (s, 3H), 3.90 (dd, *J* = 4.6, 8.2 Hz, 1H), 3.98–3.99 (m, 1H), 4.40 (d, *J* = 12.0 Hz, 1H), 4.69 (d, *J* = 12.0 Hz, 1H), 7.27–7.36 (m, 5H); ^13^C NMR (100 MHz, CDCl_3_) δ −5.2, −4.8, 13.7, 17.6, 18.0, 18.4, 23.0, 25.8, 27.1, 29.5, 31.0, 34.4, 34.7, 40.7, 42.1, 51.8, 53.0, 56.4, 69.4, 72.3, 78.3, 127.8, 128.0, 128.3, 137.6, 173.6; HRMS (ESI^+^) calcd for C_30_H_50_O_4_SiNa [M + Na]^+^ 525.3371, found 525.3399.

### 3.10. Methyl (2S,5R)-5-{(1R,3aR,4S,7aR)-4-[(tert-butyldimethylsilyl)oxy]-7a-methyloctahydro-1H-inden-1-yl}-2-hydroxyhexanoate *(**9**)*

To a solution of **30** (109.0 mg, 0.22 mmol) in MeOH (10 mL) and EtOAc (2 mL), we added 10% Pd/C catalyst (22.6 mg). The mixture was stirred for 45 h at room temperature, and then for 68 h at 50 °C, under a hydrogen atmosphere. The reaction mixture was diluted with AcOEt, filtered through a Celite pad, and concentrated under reduced pressure. Purification via flash column chromatography on silica gel (hexane:EtOAc = 3:1) yielded **9** (78.0 mg, 87%) as a colorless oil.

**9**: [α]_D_^27^ +44.8 (c 1.67, CHCl_3_); IR (neat) 3488, 1742, 1461, 1370, 1257, 1081, 1020, 840, 776, 686 cm^−1^; ^1^H NMR (600 MHz, CDCl_3_) δ −0.01 (s, 3H), 0.00 (s, 3H), 0.88–0.90 (m, 15H), 0.98–1.57 (m, 13H), 1.64–1.70 (m, 1H), 1.75–1.88 (m, 3H), 1.92–1.95 (m, 1H), 2.14 (brs, 1H), 3.78 (s, 3H), 3.99–4.00 (m, 1H), 4.15 (dd, *J* = 3.9, 6.9 Hz, 1H); ^13^C NMR (150 MHz, CDCl_3_) δ −5.2, −4.8, 13.7, 17.6, 18.0, 18.6, 23.0, 25.8, 27.2, 30.6, 31.1, 34.4, 35.0, 40.7, 42.1, 52.4, 53.0, 56.4, 69.4, 71.0, 175.9; HRMS (ESI^+^) calcd for C_23_H_44_O_4_SiNa [M + Na]^+^ 435.2901, found 435.2897.

### 3.11. Methyl (2R,5R)-5-{(1R,3aR,4S,7aR)-4-[(tert-butyldimethylsilyl)oxy]-7a-methyloctahydro-1H-inden-1-yl}-2-hydroxyhexanoate *(**10**)*

To a solution of **31** (219.1 mg, 0.44 mmol) in isopropanol (10 mL), we added 10% Pd/C catalyst (62.9 mg). The mixture was stirred for 45 h at room temperature, and then for 68 h at 50 °C, under a hydrogen atmosphere. The reaction mixture was diluted with EtOAc, filtered through a Celite pad, and concentrated under reduced pressure. Purification via flash column chromatography on silica gel (hexane:EtOAc = 3:1) yielded **10** (136.5 mg, 76%) as a colorless oil.

**10**: [α]_D_^27^ +33.2 (c 0.61, CHCl_3_); IR (neat) 3506, 1739, 1468, 1253, 1085, 1025, 838, 778 cm^−1^; ^1^H NMR (600 MHz, CDCl_3_) δ −0.01 (s, 3H), 0.00 (s, 3H), 0.88–0.90 (m, 15H), 1.00–1.12 (m, 2H), 1.20–1.27 (m, 2H), 1.30–1.45 (m, 3H), 1.49–1.58 (m, 2H), 1.65–1.71 (m, 2H), 1.75–1.83 (m, 2H), 1.92–1.95 (m, 1H), 3.78 (s, 3H), 3.99–3.99 (m, 1H), 4.17–4.18 (dd, *J* = 5.6, 6.0 Hz, 1H); ^13^C NMR (150 MHz, CDCl_3_) δ −5.2, −4.8, 13.7, 17.7, 18.0, 18.5, 23.0, 25.8, 27.2, 30.5, 30.9, 34.4, 34.8, 40.7, 42.1, 52.4, 53.0, 56.4, 69.4, 70.7, 175.9; HRMS (ESI^+^) calcd for C_23_H_44_O_4_Si [M + Na]^+^ 435.2901, found 435.2887.

### 3.12. Methyl (2S,5R)-5-{(1R,3aR,4S,7aR)-4-[(tert-butyldimethylsilyl)oxy]-7a-methyloctahydro-1H-inden-1-yl}-2-fluorohexanoate *(**11**)*

DAST (48.0 mg, 43 μL, 0.30 mmol) was added to a solution of **10** (20.5 mg, 0.05 mmol) in CH_2_Cl_2_ (5 mL) at 0 °C, and the mixture was stirred at the same temperature for 90 min. After the reaction was quenched with MeOH, H_2_O, and saturated aqueous NaHCO_3_ at 0 °C, the mixture was extracted with CH_2_Cl_2_ three times, dried over Na_2_SO_4_, filtered, and concentrated. The residue was purified via flash column chromatography on silica gel (hexane:EtOAc = 8:1) to obtain **11** (15.5 mg, 75%) as a colorless oil.

**11**: [α]_D_^27^ +34.4 (c 1.03, CHCl_3_); IR (neat) 1766, 1746, 1469, 1442, 1378, 1254, 1212, 1089, 1024, 836, 776 cm^−1^; ^1^H NMR (400 MHz, CDCl_3_) δ −0.02 (s, 3H), 0.00 (s, 3H), 0.88–0.90 (m, 15H), 1.00–2.03 (m, 19H), 3.78 (s, 3H), 3.98–3.99 (m, 1H), 4.85 (ddd, *J* = 4.1, 7.3, 49.0 Hz, 1H); ^13^C NMR (100 MHz, CDCl_3_) δ −5.2, −4.8, 13.7, 17.6, 18.0, 18.4, 23.0, 25.8, 27.1, 29.1 (d, *J* = 20.0 Hz), 30.3 (d, *J* = 2.9 Hz), 34.4, 34.9, 40.7, 42.1, 52.2, 53.0, 56.3, 69.4, 89.6 (d, *J* = 183.1 Hz), 170.5 (d, *J* = 23.8 Hz); HRMS (ESI^+^) calcd for C_23_H_43_O_3_FSiNa [M + Na]^+^ 437.2858, found 437.2869.

### 3.13. Methyl (2R,5R)-5-{(1R,3aR,4S,7aR)-4-[(tert-butyldimethylsilyl)oxy]-7a-methyloctahydro-1H-inden-1-yl}-2-fluorohexanoate *(**12**)*

DAST (195.0 mg, 173 μL, 1.21 mmol) was added to a solution of **9** (99.7 mg, 0.24 mmol) in CH_2_Cl_2_ (3 mL) at 0 °C, and the mixture was stirred at the same temperature for 2 h 15 min. After the reaction was quenched with MeOH, H_2_O, and saturated aqueous NaHCO_3_ at 0 °C, the mixture was extracted with CH_2_Cl_2_ three times, dried over Na_2_SO_4_, filtered, and concentrated. The residue was purified via flash column chromatography on silica gel (hexane:EtOAc = 8:1) to obtain **12** (31.0 mg, 31%) as a colorless oil.

**12**: [α]_D_^27^ +44.7 (c 2.39, CHCl_3_); IR (neat) 1769, 1746, 1465, 1445, 1370, 1254, 1208, 1081, 1024, 836, 769 cm^−1^; ^1^H NMR (600 MHz, CDCl_3_) δ −0.01 (s, 3H), 0.00 (s, 3 H), 0.88–0.90 (m, 15H), 0.99–1.12 (m, 2H), 1.15–1.26 (m, 3H), 1.29–1.37 (m, 3H), 1.39–1.48 (m, 1H), 1.51–1.59 (m, 2H), 1.64–1.68 (m, 1H), 1.73–1.84 (m, 3H), 1.73–1.84 (m, 3H), 1.86–1.95 (m, 2H), 3.79 (s, 3H), 3.99–3.99 (m, 1H), 4.89 (ddd, *J* = 4.2, 8.4, 49.8 Hz, 1H); ^13^C NMR (150 MHz, CDCl_3_) δ −5.2, −4.8, 13.7, 17.6, 18.0, 18.4, 23.0, 25.8, 27.1, 29.1 (d, *J* = 20.1 Hz), 30.1, 34.4, 34.6, 40.7, 42.1, 52.2, 53.0, 56.3, 69.4, 89.3 (d, *J* = 182.4 Hz), 170.6 (d, *J* = 24.5 Hz); HRMS (ESI^+^) calcd for C_23_H_43_O_3_FSiNa [M + Na]^+^ 437.2858, found 437.2874.

### 3.14. (1R,3aR,4S,7aR)-1-[(2R,5S)-5-Fluoro-6-hydroxy-6-methylheptan-2-yl]-7a-methyloctahydro-1H-inden-4-ol *(**15**)*

To a solution of **11** (82.2 mg, 0.20 mmol) in THF (3 mL), we added MeMgCl (264 μL, 3.0 M THF solution, 0.79 mmol) at 0 °C, and the mixture was stirred at 0 °C for 10 min. MeMgCl (264 μL, 3.0 M THF solution, 0.79 mmol) was added to the mixture at 0 °C and stirred at the same temperature for 5 min. After the reaction was quenched with H_2_O, the mixture was extracted with EtOAc three times, washed with saturated aqueous NH_4_Cl, dried over Na_2_SO_4_, filtered, and concentrated. The crude residue was used for the next reaction without further purification. To the above crude residue in MeOH (10 mL) and CH_2_Cl_2_ (5 mL), we added *p*-toluenesulfonic acid monohydrate (399.2 mg, 2.10 mmol), and the mixture was stirred at room temperature for 24 h under air. After the reaction was quenched with H_2_O and saturated aqueous NaHCO_3_ at room temperature, the mixture was extracted with CH_2_Cl_2_ three times, dried over Na_2_SO_4_, filtered, and concentrated. The residue was purified via flash column chromatography on silica gel (hexane:EtOAc = 1:1) to obtain **15** (36.3 mg, 61%, in 2 steps) as a white powder.

**15**: [α]_D_^27^ +17.5 (c 1.30, CHCl_3_); IR (neat) 3412, 1465, 1378, 1250, 1168, 1066, 990, 731 cm^−1^; ^1^H NMR (600 MHz, CDCl_3_) δ 0.92 (d, *J* = 6.0 Hz, 3H), 0.94 (s, 3H), 1.03–1.17 (m, 3H), 1.20–1.21 (m, 6H), 1.29–1.36 (m, 2H), 1.42–1.90 (m, 13H), 1.98–2.01 (m, 1H), 4.07–4.08 (m, 1H), 4.14 (ddd, *J* = 1.8, 10.2, 48.0 Hz, 1H); ^13^C NMR (150 MHz, CDCl_3_) δ 13.5, 17.4, 18.5, 22.5, 24.3 (d, *J* = 4.4 Hz), 25.3 (d, *J* = 4.4 Hz), 26.4 (d, *J* = 21.6 Hz), 27.1, 32.1, 33.6, 35.3, 40.4, 41.9, 52.6, 56.5, 69.4, 72.0 (d, *J* = 20.1 Hz), 100.7 (d, *J* = 172.4 Hz); HRMS (ESI^−^) calcd for C_18_H_32_O_2_FSi [M-H]^−^ 299.2392, found 299.2388.

### 3.15. (1R,3aR,4S,7aR)-1-[(2R,5R)-5-Fluoro-6-hydroxy-6-methylheptan-2-yl]-7a-methyloctahydro-1H-inden-4-ol *(**16**)*

To a solution of **12** (31.0 mg, 0.075 mmol) in THF (1 mL), we added MeMgCl (150 μL, 3.0 M THF solution, 0.45 mmol) at 0 °C, and the mixture was stirred at 0 °C for 10 min. MeMgCl (264 μL, 3.0 M THF solution, 0.79 mmol) was added to the mixture at 0 °C and further stirred for 10 min. After the reaction was quenched with H_2_O, the mixture was extracted with EtOAc three times, washed with saturated aqueous NH_4_Cl, dried over Na_2_SO_4_, filtered, and concentrated. The crude residue was used for the next reaction without further purification. To the above crude residue in MeOH (10 mL) and CH_2_Cl_2_ (5 mL), we added *p*-toluenesulfonic acid monohydrate (380.7 mg, 2.0 mmol), and the mixture was stirred at room temperature for 24 h under air. After the reaction was quenched with H_2_O and saturated aqueous NaHCO_3_ at room temperature, the mixture was extracted with CH_2_Cl_2_ three times, dried over Na_2_SO_4_, filtered, and concentrated. The residue was purified via flash column chromatography on silica gel (hexane:EtOAc = 2:1) to obtain **16** (18.8 mg, 83%, in 2 steps) as a white powder.

**16**: [α]_D_^27^ +43.4 (c 1.45, CHCl_3_); IR (neat) 3402, 1469, 1374, 1168, 1073, 994 cm^−1^; ^1^H NMR (600 MHz, CDCl_3_) δ 0.91 (d, *J* = 6.0 Hz, 3H), 0.94 (s, 3H), 1.01–1.74 (m, 21H), 1.78–1.91 (m, 3H), 1.98–2.00 (m, 1H), 4.07–4.07 (m, 1H), 4.18 (ddd, *J* = 2.1, 10.5, 48.6 Hz, 1H); ^13^C NMR (150 MHz, CDCl_3_) δ 13.5, 17.4, 18.3, 22.5, 24.2 (d, *J* = 4.4 Hz), 25.4 (d, *J* = 4.4 Hz), 26.0 (d, *J* = 21.6 Hz), 27.1, 31.7, 33.6, 34.9, 40.4, 41.9, 52.6, 56.4, 69.4, 72.0 (d, *J* = 20.1 Hz), 99.9 (d, *J* = 170.9 Hz); HRMS (APCI^−^) calcd for C_18_H_33_O_2_FSi [M-H]^−^ 299.2392, found 299.2420.

### 3.16. (24S)-24-Fluoro-25-hydroxyvitamin D_3_
*(**3**)*

4-Methylmorpholine *N*-oxide (32.6 mg, 0.28 mmol) was added to a solution of **15** (22.2 mg, 0.074 mmol) in CH_2_Cl_2_ (2 mL), and the mixture was cooled to 0 °C. Tetrapropylammonium perruthenate (TPAP, 15.2 mg, 0.043 mmol) was added to the mixture, and the mixture was stirred at room temperature for 1 h. The reaction was diluted with Et_2_O, and the mixture was directly purified via flash column chromatography on silica gel (Et_2_O only) to obtain the crude ketone, which was used for the next reaction without further purification.

TMSCl (80.4 mg, 94 μL, 0.74 mmol) was added to the 0 °C cooled solution of crude ketone and imidazole (66.5 mg, 0.98 mmol) in CH_2_Cl_2_ (2 mL), and the mixture was stirred at the same temperature for 15 min. After the reaction was quenched with H_2_O and saturated aqueous NH_4_Cl at 0 °C, the mixture was extracted with CH_2_Cl_2_ three times, dried over Na_2_SO_4_, filtered, and concentrated. The residue was purified via flash column chromatography on silica gel (hexane:EtOAc = 10:1) to obtain crude **32**.

*n*BuLi (191 μL, 1.55 M hexane solution, 0.30 mmol) was added to a solution of A-ring phosphine oxide [16] (132.6 mg, 0.29 mmol) in THF (1.5 mL) at −78 °C. After stirring for 15 min, the solution of crude **32** in THF (2 mL) was added, and the mixture was stirred at −78 °C for 15 min and 0 °C for 5 min. After the reaction was quenched with H_2_O and saturated aqueous NH_4_Cl at the same temperature, the mixture was extracted with EtOAc three times, washed with brine, dried over Na_2_SO_4_, filtered, and concentrated. The residue was purified via flash column chromatography on silica gel (hexane:EtOAc = 10:1) to obtain the crude coupling product (34.9 mg), which was used for the next reaction without further purification. Tetrabutylammonium fluoride (370 μL, 1 M THF solution, 0.37 mmol) was added to a solution of the crude coupling product (34.9 mg) in THF (2 mL), and the mixture was stirred at room temperature for 16 h. After the reaction was quenched with H_2_O at room temperature, the mixture was extracted with EtOAc three times, dried over Na_2_SO_4_, filtered, and concentrated. The residue was purified via flash column chromatography on silica gel (hexane:EtOAc = 1:1) to obtain **3** (15.6 mg, 50%, in 4 steps) as a white powder.

**3**: [α]_D_^27^ +85.1 (c 1.20, EtOH); IR (neat) 3369, 1455, 1375, 1168, 1054, 893 cm^−1^; ^1^H NMR (400 MHz, CD_3_OD) δ 0.61 (s, 3H), 1.02 (d, *J* = 6.4 Hz, 3H), 1.11–1.19 (m, 1H), 1.21 (s, 3H), 1.23 (d, *J* = 1.8 Hz, 3H), 1.35–2.26 (m, 21H), 2.45 (dt, *J* = 4.8, 13.8 Hz, 1H), 2.58 (dd, *J* = 3.7, 12.8 Hz, 1H), 2.89–2.92 (m, 1H), 3.77–3.84 (m, 1H), 4.03–4.18 (m, 1H), 4.79 (d, *J* = 1.8 Hz, 1H), 5.08 (brs, 1H), 6.08 (d, *J* = 11.4 Hz, 1H), 6.26 (d, *J* = 11.4 Hz, 1H); ^13^C NMR (100 MHz, CD_3_OD) δ 12.7, 19.7, 23.6, 24.9, 25.0 (d, *J* = 2.9 Hz), 26.0 (d, *J* = 2.8 Hz), 27.5, 27.8, 29.0, 30.2, 33,9, 34.0, 36.9, 37.8, 42.2, 47.3 (d, *J* = 13.4 Hz), 57.8, 58.1, 70.9, 72.7 (d, *J* = 21.0 Hz), 101.6 (d, *J* = 173.5 Hz), 113.0, 119.3, 122.9, 137.6, 142.8, 147.3; HRMS (ESI^+^) calcd for C_27_H_43_O_2_FNa [M + Na]^+^ 441.3139, found 441.3106.

### 3.17. (24R)-24-Fluoro-25-hydroxyvitamin D_3_
*(**4**)*

4-Methylmorpholine *N*-oxide (26.1 mg, 0.22 mmol) was added to a solution of **16** (18.8 mg, 0.063 mmol) in CH_2_Cl_2_ (2 mL), and the mixture was cooled to 0 °C. TPAP (11.2 mg, 0.032 mmol) was added to the mixture, and the mixture was stirred at 0 °C for 10 min and room temperature for 20 min. The reaction was diluted with Et_2_O, and the mixture was directly purified via flash column chromatography on silica gel (Et_2_O only) to obtain the crude ketone, which was used for the next reaction without further purification.

TMSCl (68.4 mg, 80 μL, 0.63 mmol) was added to the 0 °C cooled solution of crude ketone and imidazole (43.7 mg, 0.64 mmol) in CH_2_Cl_2_ (2 mL), and the mixture was stirred for 7 min at room temperature. After the reaction was quenched with H_2_O and saturated aqueous NH_4_Cl, the mixture was extracted with CH_2_Cl_2_ three times, dried over Na_2_SO_4_, filtered, and concentrated. The residue was purified via flash column chromatography on silica gel (hexane:EtOAc = 10:1) to obtain crude **33**.

*n*BuLi (163 μL, 1.55 M hexane solution, 0.25 mmol) was added to a solution of A-ring phosphine oxide [16] (117.4 mg, 0.26 mmol) in THF (1.5 mL) at −78 °C. After stirring for 15 min, a solution of crude **33** in THF (2 mL) was added, and the mixture was stirred at −78 °C for 15 min and 0 °C for 5 min. After the reaction was quenched with H_2_O at the same temperature, the mixture was extracted with EtOAc three times, washed with brine, dried over Na_2_SO_4_, filtered, and concentrated. The residue was purified via flash column chromatography on silica gel (hexane:EtOAc = 10:1) to obtain the crude coupling product (24.7 mg), and it was used for the next reaction without further purification. Tetrabutylammonium fluoride (315 μL, 1 M THF solution, 0.32 mmol) was added to a solution of the crude coupling product (24.7 mg) in THF (2 mL), and the mixture was stirred at room temperature for 16 h. After the reaction was quenched with H_2_O at room temperature, the mixture was extracted with EtOAc three times, dried over MgSO_4_, filtered, and concentrated. The residue was purified via flash column chromatography on silica gel (hexane:EtOAc = 1:1) to obtain **4** (16.0 mg, 61%, in 4 steps) as a white powder.

**4**: [α]_D_^27^ +84.2 (c 1.24, EtOH); IR (neat) 3381, 1455, 1375, 1168, 1054, 881 cm^−1^; ^1^H NMR (600 MHz, CD_3_OD) δ 0.62 (s, 3H), 1.02 (d, *J* = 6.6 Hz, 3H), 1.22 (d, *J* = 1.2 Hz, 3H), 1.23 (d, *J* = 1.2 Hz, 3H), 1.34–1.42 (m, 4H), 1.48–1.77 (m, 10H), 1.94–2.09 (m, 4H), 2.14–2.25 (m, 2H), 2.45 (dt, *J* = 5.1, 13.8 Hz, 1H), 2.58 (dd, *J* = 3.9, 12.6 Hz, 1H), 2.89–2.92 (m, 1H), 3.79–3.83 (m, 1H), 4.15 (ddd, *J* = 1.5, 10.8, 48.6 Hz, 1H), 4.79 (d, *J* = 1.2 Hz, 1H), 5.08 (brs, 1H), 6.09 (d, *J* = 11.1 Hz, 1H), 6.27 (d, *J* = 11.1 Hz, 1H); ^13^C NMR (150 MHz, CD_3_OD) δ 12.7, 19.5, 23.5, 24.9, 25.2 (d, *J* = 2.9 Hz), 25.9 (d, *J* = 2.9 Hz), 27.2, 27.3, 29.0, 30.2, 33,5, 33.9, 36.9, 37.3, 42.2, 47.3 (d, *J* = 17.3 Hz), 57.8, 58.1, 70.9, 72.7 (d, *J* = 20.1 Hz), 100.7 (d, *J* = 173.7 Hz), 112.9, 119.3, 122.9, 137.7, 142.8, 147.3; HRMS (ESI+) calcd for C_27_H_43_O_2_FNa [M + Na]^+^ 441.3139, found 441.3133.

### 3.18. (3S,6R)-6-{(1R,3aR,4S,7aR)-4-[(tert-Butyldimethylsilyl)oxy]-7a-methyloctahydro-1H-inden-1-yl}-2-methylheptane-2,3-diol *(**36**)*

MeMgCl (0.53 mL, 3.0 M THF solution, 1.59 mmol) was added to a solution of **30** (133.4 mg, 0.265 mmol) in THF (4 mL) at 0 °C, and the mixture was stirred at 0 °C for 11 min. After the reaction was quenched with H_2_O, the mixture was extracted with EtOAc three times, washed with saturated aqueous NH_4_Cl, dried over Na_2_SO_4_, filtered, and concentrated. The residue was purified via flash column chromatography on silica gel (hexane:EtOAc = 4:1) to obtain crude **34** (130.0 mg), and it was used for the next reaction without further purification.

To a solution of crude **34** (130.0 mg) in MeOH (4 mL), we added 10% Pd/C catalyst (20.0 mg). The mixture was stirred for 6 days at room temperature under a hydrogen atmosphere. The reaction mixture was diluted with EtOAc, filtered through a Celite pad, and concentrated under reduced pressure. Purification via flash column chromatography on silica gel (hexane:EtOAc = 2:1) yielded **36** (87.2 mg, 82%) as a colorless oil [25].

**36**: [α]_D_^27^ +31.9 (c 6.71, CHCl_3_); IR (neat) 3398, 1469, 1374, 1250, 1164, 1085, 1069, 1024, 832, 776, 739 cm^−1^; ^1^H NMR (600 MHz, CDCl_3_) δ −0.02 (s, 3H), −0.01 (s, 3H), 0.87–0.90 (m, 15H), 0.97–1.41 (m, 16H), 1.50–1.57 (m, 2H), 1.64–1.81 (m, 4H), 1.92–1.95 (m, 1H), 2.40 (s, 2H), 3.25 (dd, *J* = 2.1, 9.9 Hz, 1H), 3.98–3.98 (m, 1H); ^13^C NMR (150 MHz, CDCl_3_) δ −5.2, −4.8, 13.7, 17.6, 18.0, 18.7, 23.0, 23.1, 25.8, 26.5, 27.3, 28.3, 33.1, 34.4, 35.4, 40.7, 42.1, 53.0, 56.6, 69.4, 73.3, 79.6; HRMS (ESI^+^) calcd for C_24_H_48_O_3_SiNa [M + Na]^+^ 435.3265, found 435.3271.

### 3.19. (3R,6R)-6-{(1R,3aR,4S,7aR)-4-[(tert-Butyldimethylsilyl)oxy]-7a-methyloctahydro-1H-inden-1-yl}-2-methylheptane-2,3-diol *(**37**)*

MeMgCl (415 μL, 3.0 M THF solution, 1.25 mmol) was added to a solution of **31** (125.1 mg, 0.249 mmol) in THF (3 mL) at 0 °C, and the mixture was stirred at 0 °C for 7 min. After the reaction was quenched with H_2_O, the mixture was extracted with EtOAc three times, washed with saturated aqueous NH_4_Cl, dried over Na_2_SO_4_, filtered, and concentrated. The residue was purified via flash column chromatography on silica gel (hexane:EtOAc = 4:1) to obtain crude **35**, which was used for the next reaction without further purification.

To a solution of crude **35** in MeOH (4 mL), we added 10% Pd/C catalyst (20.0 mg). The mixture was stirred for 68 h at room temperature under a hydrogen atmosphere. The reaction mixture was diluted with EtOAc, filtered through a Celite pad, and concentrated under reduced pressure. Purification via flash column chromatography on silica gel (hexane:EtOAc = 2:1) yielded **37** (24.1 mg, 23%, **35** recovery 54%) as a colorless oil [25].

**37**: [α]_D_^27^ +62.5 (c 1.85, CHCl_3_); IR (neat) 3409, 1469, 1378, 1254, 1164, 1073, 1024, 840, 772, 739 cm^−1^; ^1^H NMR (600 MHz, CDCl_3_) δ −0.01 (s, 3H), 0.00 (s, 3H), 0.88–0.91 (m, 15H), 0.98–1.48 (m, 18H), 1.51–1.58 (m, 1H), 1.65–1.67 (m, 1H), 1.76–1.86 (m, 2H), 1.93–1.96 (m, 4H), 3.32–3.34 (m, 1H), 4.00–4.00 (m, 1H); ^13^C NMR (150 MHz, CDCl_3_) δ −5.2, −4.8, 13.7, 17.7, 18.0, 18.5, 23.0, 23.2, 25.8, 26.6, 27.4, 28.1, 32.7, 34.4, 35.1, 40.7, 42.1, 53.0, 56.7, 69.5, 73.2, 78.8; HRMS (ESI^+^) calcd for C_24_H_48_O_3_SiNa [M + Na]^+^ 435.3265, found 435.3282.

### 3.20. (3S,6R)-6-[(1R,3aR,4S,7aR)-4-Hydroxy-7a-methyloctahydro-1H-inden-1-yl]-2-methylheptane-2,3-diol *(**13**)*

*p*-Toluenesulfonic acid monohydrate (199.1 mg, 1.01 mmol) was added to a solution of **36** (46.5 mg, 0.11 mmol) in MeOH (4 mL) and CH_2_Cl_2_ (4 mL), and the mixture was stirred at room temperature for 45 h under air. After the reaction was quenched with H_2_O and saturated aqueous NaHCO_3_ at room temperature, the mixture was extracted with CH_2_Cl_2_ three times, washed with brine, dried over Na_2_SO_4_, filtered, and concentrated. The residue was purified via flash column chromatography on silica gel (EtOAc only) to obtain **13** (30.4 mg, 90%) as a colorless oil. The spectral data of the product matched those reported in the literature [25].

### 3.21. (3R,6R)-6-[(1R,3aR,4S,7aR)-4-Hydroxy-7a-methyloctahydro-1H-inden-1-yl]-2-methylheptane-2,3-diol *(**14**)*

*p*-Toluenesulfonic acid monohydrate (192.9 mg, 1.01 mmol) was added to a solution of **37** (49.2 mg, 0.12 mmol) in MeOH (5 mL) and CH_2_Cl_2_ (5 mL), and the mixture was stirred at room temperature for 53 h under air. After the reaction was quenched with H_2_O and saturated aqueous NaHCO_3_ at room temperature, the mixture was extracted with CH_2_Cl_2_ three times, washed with brine, dried over Na_2_SO_4_, filtered, and concentrated. The residue was purified via flash column chromatography on silica gel (EtOAc only) to obtain **14** (30.4 mg, 85%) as a colorless oil. The spectral data of the product matched those reported in the literature [25].

### 3.22. (1R,3aR,4S,7aR)-7a-Methyl-1-{(R)-4-[(S)-2,2,5,5-tetramethyl-1,3-dioxolan-4-yl]butan-2-yl}octahydro-1H-inden-4-ol *(**38**)*

PPTS (15.8 mg, 0.06 mmol) was added to the solution of **13** (30.4 mg, 0.10 mmol) in acetone (1 mL) and 2,2-dimethoxypropane (1 mL), and the mixture was stirred at room temperature for 19 h under air. After the reaction was quenched with H_2_O and saturated aqueous NaHCO_3_ at room temperature, the mixture was extracted with EtOAc three times, dried over Na_2_SO_4_, filtered, and concentrated. The residue was purified via flash column chromatography on silica gel (hexane:EtOAc = 1:1) to obtain **38** (32.3 mg, 94%) as a colorless oil. The spectral data of the product matched those reported in the literature [25].

### 3.23. (1R,3aR,4S,7aR)-7a-Methyl-1-{(R)-4-[(R)-2,2,5,5-tetramethyl-1,3-dioxolan-4-yl]butan-2-yl}octahydro-1H-inden-4-ol *(**39**)*

PPTS (19.4 mg, 0.08 mmol) was added to a solution of **14** (30.4 mg, 0.10 mmol) in acetone (1 mL) and 2,2-dimethoxypropane (1 mL), and the mixture was stirred at room temperature for 4 h under air. After the reaction was quenched with H_2_O and saturated aqueous NaHCO_3_ at room temperature, the mixture was extracted with EtOAc three times, dried over Na_2_SO_4_, filtered, and concentrated. The residue was purified via flash column chromatography on silica gel (hexane:EtOAc = 1:1) to obtain **39** (28.6 mg, 83%) as a colorless oil. The spectral data of the product matched those reported in the literature [25].

### 3.24. (1R,3aR,7aR)-7a-Methyl-1-{(R)-4-[(S)-2,2,5,5-tetramethyl-1,3-dioxolan-4-yl]butan-2-yl}octahydro-4H-inden-4-one *(**40**)*

4-Methylmorpholine *N*-oxide (31.2 mg, 0.27 mmol) was added to a solution of **38** (32.3 mg, 0.095 mmol) in CH_2_Cl_2_ (2 mL), and the mixture was cooled to 0 °C. TPAP (18.3 mg, 0.052 mmol) was added to the mixture, and the mixture was stirred at 0 °C for 1 h. The reaction was diluted with an excess amount of Et_2_O. The mixture was directly purified via flash column chromatography on silica gel (Et_2_O only), followed by purification via flash column chromatography on silica gel (hexane:EtOAc = 4:1), to obtain **40** (25.2 mg, 79%) as a colorless oil. The spectral data of the product matched those reported in the literature [25].

### 3.25. (1R,3aR,7aR)-7a-Methyl-1-{(R)-4-[(R)-2,2,5,5-tetramethyl-1,3-dioxolan-4-yl]butan-2-yl}octahydro-4H-inden-4-one *(**41**)*

4-Methylmorpholine *N*-oxide (28.8 mg, 0.25 mmol) was added to a solution of **39** (28.6 mg, 0.085 mmol) in CH_2_Cl_2_ (1 mL), and the mixture was cooled to 0 °C. TPAP (13.9 mg, 0.04 mmol) was added to the mixture, and the mixture was stirred at 0 °C for 40 min. The reaction was diluted with Et_2_O, and the mixture was directly purified via flash column chromatography on silica gel (Et_2_O only), followed by purification via flash column chromatography on silica gel (hexane:EtOAc = 4:1), to obtain **41** (28.2 mg, 99%) as a colorless oil. The spectral data of the product matched those reported in the literature [25].

### 3.26. (24S)-24,25-Dihydroxyvitamin D_3_
*(**5**)*

*n*BuLi (145 μL, 1.55 M hexane solution, 0.225 mmol) was added to a solution of A-ring phosphine oxide [16] (101.4 mg, 0.22 mmol) in THF (1 mL) at −78 °C. After stirring for 15 min, a solution of **40** (25.2 mg, 0.075 mmol) in THF (1.5 mL) was added, and the mixture was stirred at −78 °C for 2 h. After the reaction was quenched with H_2_O at the same temperature, the mixture was extracted with EtOAc three times, washed with brine, dried over Na_2_SO_4_, filtered, and concentrated. The residue was purified via flash column chromatography on silica gel (hexane:EtOAc = 10:1) to obtain the crude coupling product (39.4 mg), which was used for the next reaction without further purification. Tetrabutylammonium fluoride (414 μL, 1 M THF solution, 0.414 mmol) was added to the solution of the crude coupling product (39.4 mg) in THF (3 mL), and the mixture was stirred at room temperature for 15 h. After the reaction was quenched with H_2_O and aqueous saturated NH_4_Cl at room temperature, the mixture was extracted with EtOAc three times, dried over Na_2_SO_4_, filtered, and concentrated. The residue was purified via flash column chromatography on silica gel (hexane:EtOAc = 1:2) to obtain the crude product, which was used for the next reaction without further purification.

The above crude residue was dissolved in MeOH (10 mL), and AG 50W-X4 resin (177.2 mg) was added. The mixture was then stirred for 26 h, and the solids were filtered off, washed with MeOH, and the solution was concentrated in vacuo. The residue was purified via flash column chromatography (hexane:EtOAc = 1:2) to obtain **5** (20.7 mg, 66%) as a white powder. The spectral data of the product matched those reported in the literature [25].

### 3.27. (24R)-24,25-Dihydroxyvitamin D_3_
*(**6**)*

*n*BuLi (163 μL, 1.55 M hexane solution, 0.252 mmol) was added to a solution of A-ring phosphine oxide [16] (110.7 mg, 0.24 mmol) in THF (1 mL) at −78 °C. After stirring for 20 min, a solution of **41** (28.2 mg, 0.084 mmol) in THF (1 mL) was added, and the mixture was stirred at −78 °C for 2 h 30 min. After the reaction was quenched with H_2_O at the same temperature, the mixture was extracted with EtOAc three times, washed with brine, dried over Na_2_SO_4_, filtered, and concentrated. The residue was purified via flash column chromatography on silica gel (hexane:EtOAc = 10:1) to obtain the crude coupling product (43.3 mg), which was used for the next reaction without further purification. Tetrabutylammonium fluoride (420 μL, 1 M THF solution, 0.42 mmol) was added to the solution of the crude coupling product (43.3 mg) in THF (3 mL), and the mixture was stirred at room temperature for 17 h. After the reaction was quenched with H_2_O and aqueous saturated NH_4_Cl at room temperature, the mixture was extracted with EtOAc three times, dried over Na_2_SO_4_, filtered, and concentrated. The residue was purified via flash column chromatography on silica gel (hexane:EtOAc = 1:1) to obtain the crude product, which was used for the next reaction without further purification.

The above crude residue was dissolved in MeOH (5 mL), and AG 50W-X4 resin (167.5 mg) was added. The mixture was stirred for 24 h, and the solids were filtered off, washed with MeOH, and the solution was concentrated in vacuo. The residue was purified via flash column chromatography (hexane:EtOAc = 1:2) to obtain **6** (26.6 mg, 76%, in 3 steps) as a white powder. The spectral data of the product matched those reported in the literature [25].

### 3.28. Measurement of the hVDR Binding Affinity of ***3***, ***4***, and 24,24-Difluoro-25(OH)D_3_

The binding affinity of each analogue for hVDR was evaluated using an in vitro system based on the split-luciferase technique described in our previous study [29]. Briefly, 50 μL of cell lysate prepared from recombinant *Escherichia coli* expressing split-luciferase vitamin D biosensor protein [29] was added to each well of a 96-well plate, and left for 10 min at room temperature. Then, 50 μL of the luciferin solution containing 20 mM MgSO_4_, 2 mM D-luciferin, and 4 mM adenosine triphosphate in 25 mM Tris-HCl (pH 7.4) was injected into each well and incubated for 15 min at room temperature. The luminescence (photon counts) was measured using a luminometer. The relative hVDR binding affinity of each analogue was evaluated based on the concentration at which the luminescence showed 50% of the maximum value.

### 3.29. Metabolism of 25(OH)D_3_ and Its Analogues by Recombinant hCYP24A1

The metabolism of 25(OH)D_3_ and its analogues **3** and **4** by CYP24A1 was analyzed using the membrane fraction prepared from the recombinant *Escherichia coli* cells expressing human CYP24A1, as described in our previous study [30]. Briefly, the reaction mixture containing 0.02 µM human CYP24A1, 2.0 µM adrenodoxin (ADX), 0.2 µM NADPH-adrenodoxin reductase (ADR), 1 mM EDTA, 1 mM NADPH, and 5.0 µM of each substrate in 100 mM Tris-HCl (pH 7.4) was incubated at 37 °C for 5 or 15 min. The metabolites were extracted with 4 volumes of CHCl_3_-CH_3_OH (3:1) and analyzed via HPLC under the following conditions: column, CAPCELL PAK C18 UG120 (5 μm) (4.6 mm × 250 mm) (SHISEIDO, Tokyo, Japan); UV detection, 265 nm; flow rate, 1.0 mL min^−1^; column temperature, 40 °C; mobile phase, CH_3_CN: a linear gradient of 20–100% CH_3_CN aqueous solution per 25 min and 100% CH_3_CN for 10 min.

## 4. Conclusions

In summary, in this paper we described novel stereoselective syntheses of 24-fluoro-25-hydroxyvitamin D_3_ (**3** and **4**) and 24,25-dihydroxyvitamin D_3_ (**5** and **6**). To our knowledge, this is the first reported study to synthesize both 24*R-* and 24*S*-24-fluorinated vitamin D_3_ analogues. This approach also provides a practical synthetic route to one of the main natural metabolites of 25(OH)D_3_ by hCYP24A1—(24*R*)-24,25-dihydroxyvitamin D_3_ (**6**). This synthetic method paves the way for efficient access to 24-substituted vitamin D_3_ analogues. Synthesis of new 24-substituted vitamin D_3_ analogues utilizing this method, along with evaluation of their biological activities, is in progress.

## Data Availability

The data presented in this study are available on request from the corresponding author.

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
