# Peer review of "Stereoselective Synthesis of 24-Fluoro-25-Hydroxyvitamin D3 Analogues and Their Stability to hCYP24A1-Dependent Catabolism"

_ijms, 2021, doi:10.3390/ijms222111863_

Round 1

Reviewer 1 Report

This manuscript describes convergent syntheses of two 24-fluorinated derivatives of 25-hydroxyvitamin D3 as well as their 24-hydroxylated analogs being the known metabolites of the parent 25-OH-D3. The elaborated synthetic path can be easily applied to the stereoselective preparation of other side-chain-fluorinated steroids of potential pharmaceutical application.

Taking these facts into consideration this reviewer recommends it for publication in the International Journal of Molecular Sciences if authors revise the manuscript according to the comments shown in the attached file.

Author Response

Thank you very much for checking our manuscript in detail, and for positive comments for publication of the revised manuscript. We revised our manuscript according to the comments as shown in the attached file one by one, and our opinions to the selected comments from reviewer 1:

line 170: we often use this description of “flash column chromatography”.

line 308: we re-investigated HR-MS of compound 29, and it was 487.3269.

lines 482, 520, 659, and 681: in oil, it described as colorless; but in powder, it is white when it has no color.

Other comments of reviewer 1 is reflected in the revised manuscript.

Reviewer 2 Report

The authors presented a very interesting work on the synthesis of fluorinated derivatives of vitamin D. The work is done at a good level and is very easy to read. It is very pleasant to read such papers. On the whole, the paper makes a positive impression. However, there are several comments on this work. The main purpose of the synthesis of fluorinated derivatives is not clear, namely to make vitamin D more slowly or not at all metabolized in the body? If so, to achieve what effect from the point of view of physiology? In the title of the paper, the authors write “… Evaluation of their preliminary biological activity”. However, the text does not contain information about their biological activity. There are only data on Relative hVDR binding affinity of 24-fluorinated 25 (OH) D3 (Table 1) and data on metabolism (Table 2), which does not correspond to the title of the paper. Authors should carefully review the article. The question also arises regarding the isomerism of compounds 3-6 with respect to the double bond at the C-ring. Is it really only one isomer that is formed and not a mixture of E and Z Isomers? If only one, then why, and how was the isomer confirmed? The authors should provide graphical NMR spectra, including two-dimensional ones, elemental analysis data for all new compounds, and also indicate the melting points for crystalline compounds. After corrections, the article can be accepted

Author Response

The authors presented a very interesting work on the synthesis of fluorinated derivatives of vitamin D. The work is done at a good level and is very easy to read. It is very pleasant to read such papers. On the whole, the paper makes a positive impression. However, there are several comments on this work. The main purpose of the synthesis of fluorinated derivatives is not clear, namely to make vitamin D more slowly or not at all metabolized in the body? If so, to achieve what effect from the point of view of physiology?

Thank you for the comments.

We have experienced that importance of half-life time of vitamin D analogues during in vivo studies. In vivo potency of drug candidates is not correlate to results from in vitro experiments, and that is often caused by short half-life of the compounds in vivo. Our recent papers on this regard: 1) effect of vitamin D side-chain fluorination on in vivo activity, see, Kawagoe, F. et al. "Discovery of a Vitamin D Receptor-Silent Vitamin D Derivative That Impairs Sterol Regulatory Element-Binding Protein In Vivo" J. Med. Chem. 2021, 64 (9), 5689-5709.

According to reviewer’s above comments, we revised the manuscript in p.2 at the last sentence as follows: Encouraged by these results, we have been interested in 24-substituted vitamin D3 analogs, 24-fluoro-25-hydroxyvitamin D3 (3,4), to study elongation of half-life time of 25(OH)D3 against CYP24A1-dependent metabolism [11]. 

We added this reference as ref. 11 in the revised manuscript.

These are also our experience in vivo studies, 2) effect of A-ring modification with 2alpha-substitution to elongate half-life on in vivo activity, see, Matsuo, M. et al. “Synthesis of 2a-Heteroarylalkyl Active Vitamin D3 with Therapeutic Effect on Enhancing Bone Mineral Density in vivoACS Med. Chem. Lett. 2013, 4 (7), 671-674; Chiang, K.-C. et al. “Evaluation of the Potential Therapeutic Role of a New Generation of Vitamin D Analog, MART-10, in Human Pancreatic Cancer Cells in Vitro and in Vivo” Cell Cycle 2013, 12 (8), 1316-1325.

In the title of the paper, the authors write “… Evaluation of their preliminary biological activity”. However, the text does not contain information about their biological activity. There are only data on Relative hVDR binding affinity of 24-fluorinated 25 (OH) D3 (Table 1) and data on metabolism (Table 2), which does not correspond to the title of the paper. Authors should carefully review the article.

Thank you for the comments.

We revised the title as follows: “Stereoselective Synthesis of 24-Fluoro-25-Hydroxyvitamin D3 Analogues and Their Stability to hCYP24A1-dependent Catabolism”

The question also arises regarding the isomerism of compounds 3-6 with respect to the double bond at the C-ring. Is it really only one isomer that is formed and not a mixture of E and Z Isomers? If only one, then why, and how was the isomer confirmed? The authors should provide graphical NMR spectra, including two-dimensional ones, elemental analysis data for all new compounds, and also indicate the melting points for crystalline compounds. After corrections, the article can be accepted.

Thank you for the comments.

We used well known traditional Wittig-Horner vitamin D coupling process between CD-ring-8-oxo-fragments and A-ring phosphine oxide, which undergoes stereo-selectively to yield the natural triene-25(OH)D3 configuration. We prepared and provided a Supporting Information file consisting of 1H- and 13C-NMR spectra charts of all new compounds.

Round 2

Reviewer 2 Report

The authors have improved the article according to the comments. The paper can be accepted.